# Combination Vaccines of *Fasciola gigantica* Saposin-like Protein-2 and Leucine Aminopeptidase

**DOI:** 10.3390/tropicalmed8070334

**Published:** 2023-06-22

**Authors:** Narin Changklungmoa, Werachon Cheukamud, Wipaphorn Jaikua, Krai Meemon, Prasert Sobhon, Pornanan Kueakhai

**Affiliations:** 1Faculty of Allied Health Sciences, Burapha University, Long-Hard Bangsaen Road, Saen Sook Sub-District, Mueang District, Chonburi 20131, Thailand; narinchang@go.buu.ac.th (N.C.); werachon_ch@hotmail.com (W.C.); wipaphorn.ja@go.buu.ac.th (W.J.); 2Research Unit for Vaccine and Diagnosis of Parasitic Diseases, Burapha University, Long-Hard Bangsaen Road, Mueang District, Chonburi 20131, Thailand; 3Department of Anatomy, Faculty of Science, Mahidol University, Rama VI Rd, Bangkok 10400, Thailand; krai.mee@mahidol.ac.th (K.M.); prasertsobhon@gmail.com (P.S.)

**Keywords:** *Fasciola gigantica*, leucine aminopeptidase, saposin-like protein-2, tissue expression, vaccine potential

## Abstract

Saposin-like protein-2 (SAP-2) and leucine aminopeptidase (LAP) are major proteins involved in the digestive process of *Fasciola gigantica* (Fg). Both SAP-2 and LAP are highly expressed in *F. gigantica*; therefore, they could be vaccine candidates for fasciolosis. The aims of this study are (1) to observe the tissue expression of *F. gigantica* SAP-2 (FgSAP-2) and *F. gigantica* LAP (FgLAP) in *F. gigantica* by indirect immunofluorescence technique under confocal microscopy and (2) to test the vaccine potentials of individual and combined recombinant (r) FgSAP-2 and rFgLAP against *F. gigantica* in Imprinting Control Region (ICR) mice (n = 10 per group). By indirect immunofluorescence-confocal microscopy, FgSAP-2 and FgLAP were localized in the caecal epithelium but at different sites: FgSAP-2 appeared in small granules that are distributed in the middle and lower parts of the cytoplasm of epithelial cells, while FgLAP appeared as a line or zone in the apical cytoplasm of caecal epithelial cells. For vaccine testing, the percent protection of combined rFgSAP-2 and rFgLAP vaccines against *F. gigantica* was at 80.7 to 81.4% when compared with aluminum hydroxide (alum) adjuvant and unimmunized controls, respectively. The levels of IgG1 and IgG2a in the sera were significantly increased in single and combine vaccinated groups compared with the control groups. Vaccinated mice showed reduced liver damage when compared with control groups. This study indicates that the combined rFgSAP-2 and rFgLAP vaccine had a higher vaccine potential than a single vaccine. These results support the further testing and application of this combined vaccine against *F. gigantica* infection in farmed livestock animals.

## 1. Introduction

Fasciolosis is a zoonotic disease in which animal and human liver disease is caused by *Fasciola hepatica* or *F. gigantica* infections. The definitive hosts for *Fasciola* spp. are humans and domestic animals, including cattle, goats, sheep, and buffaloes, as well as wild animals. Fasciolosis causes economic losses regarding livestock products such as milk and meat, and significant monetary costs are also attributed to medical treatment for animals and humans. The pathogenesis of fasciolosis involves pre-hepatic and hepatic stages: the former is caused by newly excysted juveniles (NEJs) penetrating through the host’s intestinal walls and peritoneal cavity, leading to hemorrhage and inflammation, and the latter occurs during the juvenile parasites’ migration through the liver parenchyma before they reach and reside in the bile ducts, where they mature into adult stage [1,2,3]. Control strategies are designed to block the parasites’ life cycles by using anthelmintics, grazing management, and biological control. Triclabendazole is used against *Fasciola* spp. infection in hosts, but long-term use of this drug causes resistance [4,5,6,7]. Vaccination is a more cost-effective, sustainable, and environmentally friendly method for controlling the disease and is considered safe for animals and consumers [8]. Previous publications showed that several *Fasciola* antigens have been tested for the vaccine against fasciolosis, including fatty acid binding protein (FABP) [9], cathepsins B2 and B3 [10], glutathione S-transferase (GST) [11,12], cathepsin L [13,14,15,16], tetraspanin 2 (TSP2) [17], leucine aminopeptidase (LAP) [18,19,20], and saposin-like protein (SAP) [21,22,23,24,25]. Immunizations with recombinant proteins from *F***.**
*gigantica*, including 2-Cys peroxiredoxin (Prx) in mice and buffaloes [26,27], cathepsin L1 in mice, cattle, and sheep [14,15,16], cathepsin L1H in mice [13], cathepsin L1G in mice [28], GST in mice and buffalo [11,12], superoxide dismutase (SOD) in mice [29], cathepsin B2 and B3 [10], LAP in mice [18], SAP-1 in mice [21], and SAP-2 in mice [22] have been shown to protect against *F***.**
*gigantica* infection with a limited outcome. The percent protection varied due to the property of antigens themselves and the pattern of vaccine application as single or combined vaccines. *F. gigantica* expresses several proteins that mediate invasion, migration, digestion, and survival in hosts, including CatB3, CatB2, CatL1G, and CatL1H [10,13,28,30] for migration, GST, SOD, Prx, and thioredoxin glutathione reductase (TGR) for antioxidation against free radicals released by the hosts’ immune cells [11,26,29,31,32] and LAP, SAP-2, and CatL1 [33,34,35] for digestion. Therefore, an effective vaccine should aim at blocking the survival of the parasites, especially proteins that mediate these functions and are highly expressed in juvenile-to-adult parasites that have not yet fully developed immune evasion mechanisms. LAP and SAP-2 are important digestive proteins used by parasites for digesting food materials. Saposin-like proteins, a family of lipid-interacting proteins, bind to cell membranes to induce cell lysis [36]. *Fasciola* spp. use these lytic proteins to cause lysis of the hosts’ erythrocytes and leucocytes so that their contents can be digested further for the parasites’ nutrients [37]. LAP is a cytosolic metallo-exoprotease belonging to the M17 peptidase family. LAP catalyzes the breakdown of peptides at the N-terminals [38]. Because of its important role in digestion, and possibly also invasion and migration through the host’s tissues, LAP is considered a target for drugs, as well as a vaccine candidate for many parasites, particularly those causing fasciolosis [18,19,39,40,41]. Immunizations with recombinant proteins (SAP-2 and LAP) from *Fasciola* spp., including LAP in sheep [20] and SAP-2 in rabbit and mice [24,25], have been shown to confer high percents of protection against *F. hepatica* and *F. gigantica.* In previous reports, FgSAP-2 and FgLAP have been tested for a vaccine (single vaccine) in mice by using Freund’s adjuvant; the percents of protection of the FgSAP-2 vaccine and the FgLAP vaccine were estimated to be 76.4–78.5% and 60.8 and 64.3%, respectively [18,22]. However, both of them used Freund’s adjuvant, which the World Health Organization (WHO) did not accept for animal or economic animal uses. The main reason was that Freund’s adjuvant could damage host tissues and induce inflammation. Adjuvants are an important element of vaccines, as they help to enhance the hosts’ immune responses, and there are many adjuvants that had been used for vaccine testing, such as Freund’s, aluminum hydroxide (alum), and saponin Quil A. Alum adjuvant is an oil-in-water emulsion that has been licensed for human use by the U.S. Food and Drug Administration (FDA) and is safe for hosts [42,43]. Therefore, alum adjuvant is suitable for animal and human vaccine testing. The potential of single rFgSAP-2, rFgLAP, and combined rFgSAP-2 and rFgLAP vaccines with alum adjuvant against *F. gigantica* has never been tested, and in this report, we have demonstrated its high degree of protection against *F. gigantica* in mice. Consequently, we predicted that the combined rFgSAP-2 and rFgLAP vaccine with alum adjuvant may be a good, safe, and effective vaccine candidate against *F. gigantica* in farmed livestock animals.

## 2. Materials and Methods

### 2.1. Expression and Purification of Recombinant(r) FgSAP-2 and FgLAP

The rFgSAP-2-pET30b and rFgLAP-pET30b clones were constructed as previously described [33,34]. The *Escherichia coli* BL21(DE3) containing the rFgSAP-2-pET30b and rFgLAP-pET30b were incubated in 5 mL LB medium containing 100 µg/mL kanamycin and grown overnight at 37 °C with shaking at 300 rpm. The overnight culture of 500 µL was inoculated into 25 mL LB medium containing 100 µg/mL kanamycin and grown at 37 °C with shaking at 300 rpm until OD600 = 0.6. A sample of 5 mL was taken immediately for non-induced control. The clones were induced by 1 mM Isopropyl β- d -1-thiogalactopyranoside (IPTG) (Calbiochem, Merck, Darmstadt, Germany) for 4 h at 37 °C with shaking at 300 rpm. The pellet cells were resuspended with denature lysis buffer (8 M Urea, 10 mM Tris-HCl, 100 mM NaH_2_PO_4_, pH 8.0) at 5 mL per gram wet weight and stored at −80 °C. 

The rFgSAP-2 and rFgLAP were purified by nickel-nitrilotriacetic acid (Ni-NTA) affinity chromatography (Qiagen, Hilden, Germany) under denaturing conditions. The frozen pellets were thawed and mixed gently by rotary shaker at room temperature (RT) for 1 h. The lysate was centrifuged at 12,000× *g* for 30 min at RT. The supernatant was collected and filtrated by 0.45 µm filter, and then the filtrated supernatant was added to Ni-NTA column. The column was washed twice with 5 mL wash buffer (8 M Urea, 10 mM Tris-HCl, 100 mM NaH_2_PO_4_, pH 6.3). The recombinant proteins were eluted 5 times with 1 mL elution buffer (8 M Urea, 10 mM Tris-HCl, 100 mM NaH_2_PO_4_, pH 4.5) as previously described by Kueakhai et al. [34] and Changklungmoa et al. [33]. The eluted protein fractions were pooled and stored at −80 °C. The purified protein fractions were thawed and dialyzed by phosphate-buffered saline (PBS; 140 mM NaCl, 2.7 mM KCl, 10 mM Na_2_HPO_4_, 1.8 mM KH_2_PO_4_, pH 7.4), pH 7.4 at 4 °C for 48 h. The dialyzed rFgSAP-2 and rFgLAP proteins were centrifuged at 4 °C, 10,000× *g* for 30 min. The purified rFgSAP-2 and rFgLAP were kept at −20 °C and analyzed by 12.5% sodium dodecyl sulfate–polyacrylamide gel electrophoresis (SDS-PAGE) and stained with Coomassie blue. 

### 2.2. Production of Polyclonal Antibody against rFgSAP-2 and rFgLAP

A 3-month-old New Zealand white (NZW) rabbit was immunized subcutaneously with rFgSAP-2 protein with Complete Freund’s adjuvant (CFA) (Sigma–Aldrich Inc., St. Louis, MO, USA) at first immunization. Then, the second and third immunizations were immunized subcutaneously with rFgSAP-2 protein with Incomplete Freund’s adjuvant (IFA) (Sigma–Aldrich Inc., St. Louis, MO, USA) as previously reported by Kueakhai et al. [34]. Sera were collected 14 days after the third immunization and stored at −20 °C until used for further studies. The 8-week-old BALB/c mice were immunized subcutaneously with rFgLAP protein with CFA (Sigma–Aldrich Inc., St. Louis, MO, USA) at first immunization. Then, the second and third immunizations were immunized subcutaneously with rFgLAP protein with IFA (Sigma–Aldrich Inc., St. Louis, MO, USA). Sera were collected 14 days after the third immunization and stored at −20 °C until used for further studies. All animals were kept in steel cages in an air-conditioned room at 22–25 °C, with light-dark cycle of 12:12 h and 50–60% humidity following the protocol approved by the Animal Care Unit and the Committee (SCMU-ACUC) (MUSC55-003-249), Faculty of Science, Mahidol University, Bangkok, Thailand.

### 2.3. Co-Localization of FgLAP and FgSAP-2 in F. gigantica Tissues

The distributions of native FgSAP-2 and FgLAP proteins in the tissues of adult *F. gigantica* were detected by indirect immunofluorescence using the rabbit polyclonal antibody (PoAb) anti-rFgSAP-2 and mouse PoAb anti-rFgLAP. Adult *F. gigantica* were fixed in 4% paraformaldehyde (*w*/*v*) in PBS, pH 7.4, at 4 °C for 3 h. They were then dehydrated with increasing concentration of ethanol (70–100%) and finally embedded in paraffin. Paraffinized *F. gigantica* were cut into five-micrometer-thick paraffin sections, deparaffinized by xylene for 10 min, and then rehydrated in decreasing concentrations of serial alcohol from 100, 95, 90, 80, and 70% for 5 min each, rinsed in tap water for 5 min, and washed with PBS, pH 7.4 for 5 min with shaking. The sections were treated in 10 mM citrate buffer, pH 6.0 for 5 min three times in 700-watt microwave oven and washed with tap water for 5 min. The sections were then washed with PBS containing 0.1% Tween-20 (PBST) for 5 min with shaking three times and treated with 0.1% glycine in PBS (*w*/*v*) for 30 min and rinsed with PBS for 5 min with shaking. Non-specific binding was blocked with 4% bovine serum albumin (BSA) in PBS (*w*/*v*) for 30 min and incubated in rabbit PoAb against rFgSAP-2 (diluted at 1:10,000) and mouse PoAb against rFgLAP (diluted at 1:50) with PBS at 4 °C, overnight. The sections were washed to remove excess antibodies with PBST, pH 7.4, for 15 min three times and then incubated with 1:1,000 secondary antibodies (Alexa 488; goat anti-rabbit IgG and Alexa 568; goat anti-mouse IgG) (Molecular Probes, Eugene, OR, USA), diluted with PBS for 1 hr at RT in the dark, then washed by PBST for 15 min three times with shaking in the dark. After that, the sections were mounted with glycerol-PBS (9:1) on glass slides, then examined and photographed under a confocal microscope (Olympus FV1000).

### 2.4. Vaccine Testing

Sixty male, 8-week-old ICR mice were used in this study. All animals were divided into six groups: (1) non-immunized and uninfected, (2) non-immunized and infected, (3) immunized with alum adjuvant (InvivoGen, San Diego, CA, USA) and infected, (4) immunized with 50 μg rFgSAP-2 plus alum adjuvant and infected, (5) immunized with 50 μg rFgLAP plus alum adjuvant and infected, and (6) immunized with 25 μg of rFgSAP-2 and 25 μg of rFgLAP plus alum adjuvant and infected (Table 1). They were kept in steel cages in an air-conditioned room at 22–25 °C, with light-dark cycle of 12:12 h and 50–60% humidity (normal conditions) at the Institutional Animal Care and Use Committee (IACUC 28/2556), Burapha University, Chonburi, Thailand. The immunization of alum adjuvant control, rFgSAP-2, rFgLAP, and combined rFgSAP-2 and rFgLAP groups were performed three times at 2-week intervals by subcutaneous route for the first, second, and third immunizations using alum adjuvant (InvivoGen, San Diego, CA, USA). Each mouse was orally infected with 15 *F. gigantica* metacercariae at 2 weeks after third immunization. *F. gigantica* metacercariae were confirmed by using the internal transcribed spacer 1 (ITS-1) and internal transcribed spacer 2 (ITS-2) primers. Blood samples were taken at preimmunization, 2 weeks post-third immunization (infection), and 4 weeks post-infection (termination). Four weeks after infection, mice were anesthetized by CO_2_ inhalation, and their peritoneal cavities were opened and washed thoroughly with 0.85% NaCl solution. Their livers were removed, immersed in 0.85% NaCl solution, and thoroughly dissected and examined for pathological lesions and worm recoveries. The worms were collected, counted, and calculated for percent protection as follows:$$\mathrm{Percent}\;\mathrm{Protection}\;(\%)=\frac{A–B}{A}\times 100$$ “A” represents the mean worm recovery from the non-immunized-infected or immunized with alum adjuvant-infected control mice, and “B” represents the mean worm recovery from rFgSAP-2- or rFgLAP- or rFgSAP-2 + rFgLAP– vaccinated mice.

### 2.5. Determination of IgG1 and IgG2a Levels by Indirect ELISA

IgG1 and IgG2a in blood samples at preimmunization, 2 weeks post-third immunization (infection), and 4 weeks post-infection (termination) were observed by indirect ELISA. Ninety-six-microtiter plates were coated with 100 µL, 1 µg concentration of rFgSAP-2 or rFgLAP in coating buffer (15 mM Na_2_CO_3_, 35 mM NaHCO_3_, pH 9.6) 1 ml at 4 °C, O/N. The coated plates were washed three times with 0.05% PBST and nonspecific was blocked with 1% BSA in PBS 150 µL per well at RT, 1 hr. After that, coated microtiter plates were washed three times with 0.05% PBST, and then 100 µL of serial diluted sera in PBS was added and incubated for 2 h at RT. The microtiter plates were washed three times with 0.05% PBST and incubated in horseradish peroxidase (HRP)-conjugated goat anti-mouse IgG subclass (IgG1 and IgG2a) (Southern Biotech, Birmingham, AL, USA) diluted with PBS at 1:5000 and used as the secondary antibody for 1 h at RT. After that, the plates were washed three times with 0.05% PBST, and 100 µL of 3,3’,5,5’-Tetramethylbenzidine (TMB) (KPL, Gaithersburg, MD, USA) per well was added and incubated for 10 min at RT in dark. Finally, enzymatic reactions were stopped by adding 100 µL of 1 N HCl per well. The optical density (OD) values were measured at 450 nm in an automatic Titertek Multiscan spectrophotometer (Flow Laboratories, McLean, VA, USA).

### 2.6. Assessment of Liver Pathological Lesion

Four weeks after infection with *F. gigantica* metacercariae, all mice (60 mice) were anaesthetized by CO_2_ inhalation, and their livers were removed and examined for gross pathological lesions. The gross pathological lesion scores from 0 to 5 of liver damage were assigned as follows: 0-no sign of damage, 1—minor damage, 2—light lesions, 3—moderate lesions, 4—heavy lesions, 5—extensive lesions [14,44,45,46]. The gross pathological lesions were scored by two pathologists.

### 2.7. Statistical Analysis

The data were tested for significant differences between control and vaccinated groups by one-way ANOVA and Kruskal–Wallis test. Correlation between titer values of antibodies and number of worm recoveries were analyzed by one-way ANOVA of variance (Spearman’s rank correlation test) using GraphPad Prism version 7.00 for Windows, GraphPad Software, La Jolla, CA, USA (www.graphpad.com). For determining the significance of differences in the experimental data, a *p*-value less than 0.05 was considered statistically significant.

## 3. Results

### 3.1. The Purified rFgSAP-2 and rFgLAP

The purified rFgSAP-2 and rFgLAP were analyzed by 12.5% SDS-PAGE analysis. The purified rFgSAP-2 and rFgLAP bands appeared at MW 12 and 56 kDa, respectively (Figure 1).

### 3.2. Co-Localization of FgSAP-2 and FgLAP

The co-localization of FgSAP-2 and FgLAP was performed on five-micrometer-thick paraffin sections of adult *F. gigantica* using rabbit anti-rFgSAP-2 and mouse anti-rFgLAP as the primary antibodies. The results showed that SAP-2 (marked with Alexa 488-green) and LAP (marked with Alexa 568-red) were present in the caecal epithelium (Figure 2). FgSAP-2 appeared as fine granules that were localized in the middle and lower parts of the caecal epithelium, while FgLAP was stained as lines along the apical part and the surface of the epithelium. 

### 3.3. Worm Recovery

The numbers of worms recovered from the single and combined vaccinated groups showed significant differences from the alum adjuvant-infected and non-immunized-infected control groups (Table 1). The worm reductions in rFgSAP-2 + rFgLAP with alum adjuvant vaccinated group when compared to the non-immunized-infected and adjuvant-infected control groups were 80.7 and 81.4%, respectively. The combined vaccines exhibited a higher percent reduction than single vaccines without significant differences (Table 1).

### 3.4. IgG1 and IgG2a Responses

The Th2 and Th1 responses represented by IgG1 and IgG2a titer were measured in all groups of mice at three intervals: preimmunization (0 weeks), 2 weeks post-third immunization (infection; 6 weeks), and 4 weeks post-infection (termination; 10 weeks) by indirect ELISA. The rFgSAP-2 and rFgLAP−specific IgG1 and IgG2a were not detected in the sera of nonimmunized-uninfected, nonimmunized-infected, and alum adjuvant−infected control groups. The titer values of rFgSAP-2 and rFgLAP−specific IgG1 and IgG2a in the vaccinated mice were significantly increased at the times of infection (6 weeks) and termination (10 weeks) when compared to the control groups (Figure 3). On the other hand, the titer values of IgG1 and IgG2a from single and combined vaccines were not significantly different.

### 3.5. Correlation

The correlations between the numbers of recovered worms and the titer levels of IgG1 and IgG2a specific to FgSAP-2 and FgLAP were determined at the pre−immunization, 2 weeks post−third immunization (infection), and 4 weeks post−infection (termination) (Figure 4A,B). The high titer levels of IgG1 and IgG2a with low numbers of worm recoveries indicated a strong correlation in all rFgSAP-2 and rFgLAP vaccinated mice. The correlation coefficients (r) of IgG1 and IgG2a at 2 weeks post-third immunization (infection) and 4 weeks post-infection (termination) were statistically significant, indicating the correlations between the IgG1 and IgG2a levels with protection against *F. gigantica* infection. 

### 3.6. Liver Pathological Lesion 

The combined rFgSAP-2+rFgLAP-vaccinated groups showed significantly lower scores of liver lesions (*p* < 0.05) than the control groups (Table 2). In comparison, no mice in the rFgSAP-2+rFgLAP vaccinated group had scores of 3, 4, and 5 (Table 2). The percent reductions of pathological lesions in groups immunized with combined vaccine were higher than single vaccines, though the differences were not statistically significant.

## 4. Discussion

The co-localization of FgSAP-2 and FgLAP was observed by indirect immunofluorescence-confocal microscope, and FgSAP-2 and FgLAP were abundantly expressed in different sites within the caecal epithelium. FgSAP-2 appeared as small granules (green dot) distributed mainly in the middle and lower parts of the epithelial cytoplasm, while FgLAP appeared as lines along the apical part and the surface of the epithelium. Both SAP-2 and LAP were reported to be highly expressed in the caecal epithelium of *F. hepatica* and *F. gigantica* [33,34,47]. Because of their high expression and secretion, the two proteins have been considered as good candidates for vaccines.

In the present study, FgSAP-2 and FgLAP were tested as single and combined vaccines with alum adjuvant. Percent protections of single vaccines of rFgSAP-2 and rFgLAP were similar to those previously tested with Freund’s, adjuvant which showed protection at 76.4–78.5 and 60.8–64.3%, respectively [18,22]. Combined vaccines of rFgSAP-2 and rFgLAP with alum adjuvant exhibited significantly higher percent protection at 80.7−81.4%. This indicated the synergistic effects of the two antigens (rFgSAP-2 and rFgLAP) in protecting against *F. gigantica* infection. The combined vaccine of rFgSAP-2 and rFgLAP could block the functions or activities of both FgSAP-2 and FgLAP more completely than FgSAP-2 or FgLAP alone. Because LAP and SAP-2 are important digestive proteins used by parasites for digesting food materials. The activity of SAP-2 is cell lysis induction, while LAP is a cytosolic exoprotease that catalyzes the breakdown of peptides. Therefore, our study confirmed that the combined rFgSAP-2 and rFgLAP vaccine showed higher percent protection than a single vaccine (rFgSAP-2 or rFgLAP only) in mice. All vaccinated groups could induce both T-helper 1 (IgG2a) and Th2 (IgG1), with Th2 (IgG1) predominating. The mixed Th2/Th1 immune responses with Th2 predominating have been reported for several antigen vaccines, including rFhCatL1 [15,48], rFgPrx [26], rFgCatL1G [28], rFgGST26 [11], rFgCatL1H [13], rFgCatB3 and rFhCatB3 [10,49], rFgSAP-1 [21], rFgSAP-2 [22], and rFgLAP [18]. Furthermore, both single and combined vaccinated mice in this study exhibited a strong correlation between the numbers of worm recoveries and the levels of both types of antibodies (IgG1 and IgG2a) at infection and termination, which is similar to several previous studies [10,13,14]. The Th2 immune response was typically observed and characterized by the increased level of IgG1 and cytokines such as IL-4, whereas a high level of IgG2a and IFN−gamma indicated Th1 immune response [50,51,52]. Therefore, it was thought that increased antibodies from the Th2 response and cytolytic immune cells from the Th1 response synergistically reacted in killing the parasites and blocking the parasite’s digestive activities, which might eventually lead to their weakness, starvation, and finally death [51]. 

Liver damage expressed as gross pathological lesions could be evaluated by scoring by naked eyes or macroscopic examinations as reported earlier [28,46]. The scores of liver damage in single and combined rFgSAP-2 and rFgLAP vaccine groups were reduced when compared with the Alum adjuvant-infected control and nonimmunized-infected control groups. Furthermore, the combined rFgSAP-2 and rFgLAP vaccine exhibited a percent reduction of gross pathological lesions of 71.42–72.97% when compared with the alum-adjuvant-infected control and nonimmunized-infected control groups (Table 2.). Nevertheless, it is possible that both the single and combined rFgSAP-2 and rFgLAP vaccines were not able to completely block and kill all parasites, allowed some of them to penetrate the liver parenchyma, and still caused a certain degree of liver damage. In addition, parasites still could use the other digestive enzymes or proteins, i.e., FgSAP-3, FgCatB, or FgCatL isoforms, that were expressed in the caecal epithelium of parasites for their survival and digestive process. However, vaccinated groups, especially the combined rFgSAP-2 and rFgLAP group with a high percent of protection, showed a relatively low degree of liver damage. This indicated that the combined vaccine group exhibited a high percent of protection (low parasite number) and a high percent reduction of gross pathological lesions (low degree of liver damage), with high levels of antibodies to block or kill parasites, leading to reduced liver damage. 

## 5. Conclusions

In conclusion, FgSAP-2 and FgLAP were expressed in the caecal epithelium at different sites: FgSAP-2 appeared in fine granules in the middle and lower parts of the cytoplasm of epithelial cells, while FgLAP appeared as a line or zone in the apical cytoplasm of caecal epithelial cells. Our study demonstrated that the combined rFgSAP-2 − rFgLAP vaccine with alum adjuvant is more efficacious than single vaccines; the percents of protection of combined and single vaccines were 80.7 to 81.4% and 66.7 to 72.9%, respectively. The combined rFgSAP-2 and rFgLAP vaccine exhibited the reduction of liver damage when compared with the control groups at 71.42 to 72.97%. Therefore, this combined vaccine should provide high potential to reduce the *F. gigantica* infection and liver damage in farmed livestock animals. Testing the combined vaccine potential in farmed livestock animals is the subject of our current research.

## Figures and Tables

**Figure 1 tropicalmed-08-00334-f001:**
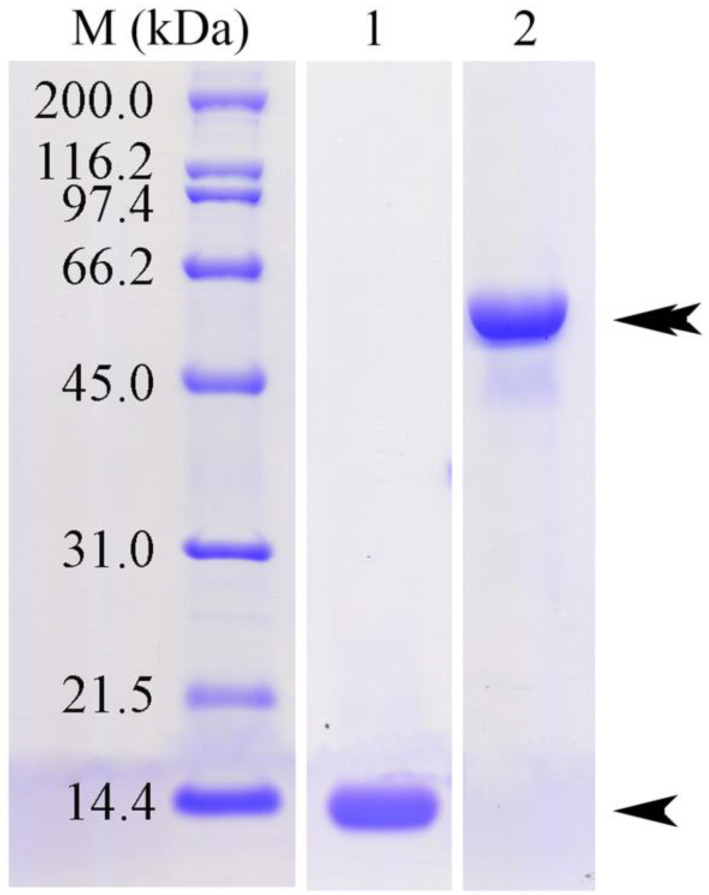
The Ni-NTA affinity purified rFgSAP-2 (lane 1) and rFgLAP (lane 2) were analyzed by 12.5% SDS-PAGE and stained with Coomassie blue. Single and double head arrows indicate MW of rFgSAP-2 and rFgLAP, respectively. Molecular weight (MW) markers (lane M) were shown on the left side.

**Figure 2 tropicalmed-08-00334-f002:**
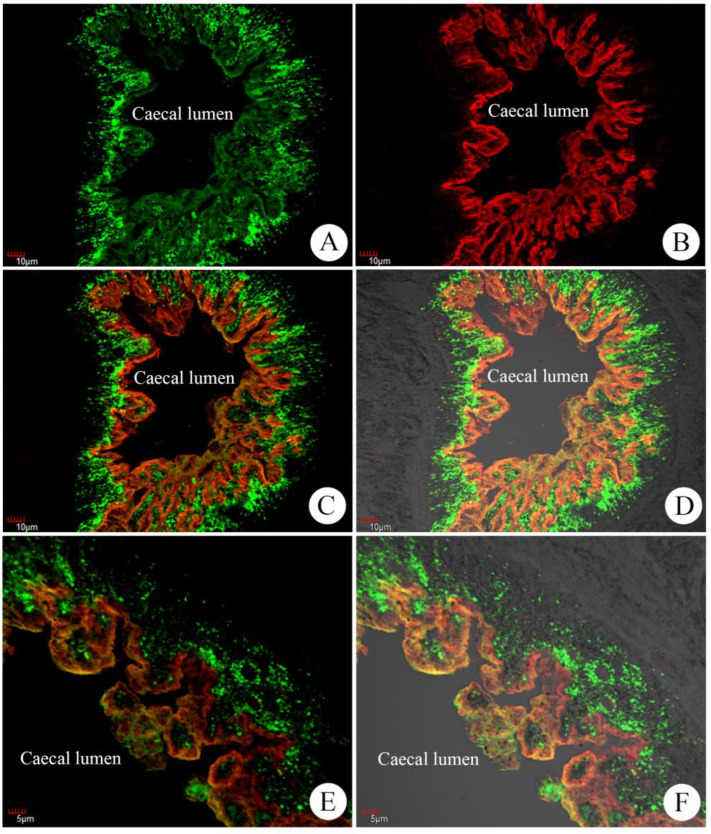
Co-localization of FgSAP-2 and FgLAP proteins in the caecal tissue of adult *F. gigantica* by indirect immunofluorescence technique, using rabbit anti-rFgSAP-2 and mouse anti-rFgLAP as probes and observed by a confocal microscope. (**A**) rabbit anti-rFgSAP-2; (**B**) mouse anti-rFgLAP; (**C**–**F**) merged pictures. FgSAP-2 (Alexa 488: green color)) and FgLAP ((Alexa 568: red color)) were localized in the different regions of caecal epithelium (**C**–**F**).

**Figure 3 tropicalmed-08-00334-f003:**
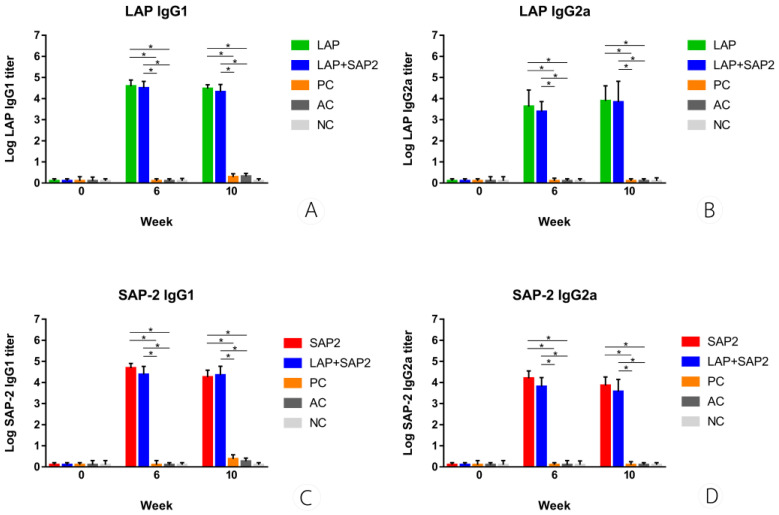
Levels of IgG1 and IgG2a specific rFgLAP (**A**,**B**), and IgG1 and IgG2a specific rFgSAP-2 (**C**,**D**) in rFgSAP-2 plus alum adjuvant and infected group (SAP-2), rFgLAP plus alum adjuvant and infected group (LAP), rFgSAP-2 + rFgLAP plus alum adjuvant and infected group (LAP+SAP-2) of mice at three intervals. * Significantly different from control groups; Negative Control (NC), Positive Control (PC), Adjuvant Control (AC), (*p*-value < 0.05).

**Figure 4 tropicalmed-08-00334-f004:**
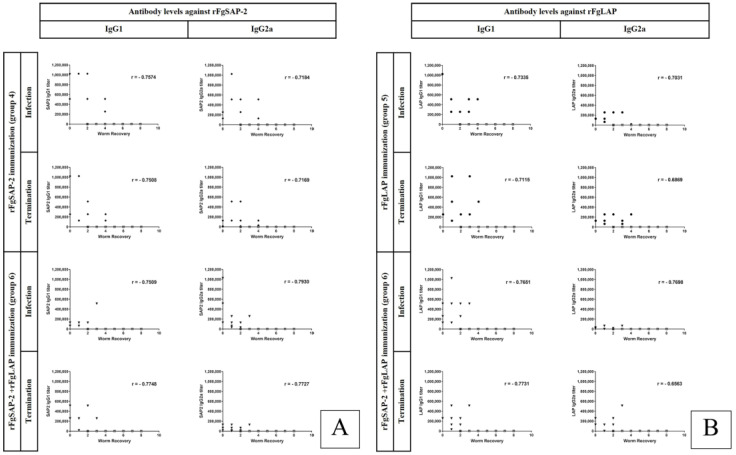
Correlations between IgG titers of IgG1, IgG2a−specific rFgSAP-2 (**A**), −specific rFgLAP (**B**) and the numbers of worm recovery at 2 weeks post-third immunization (infection) and 4 weeks post infection (termination) of the rFgLAP, rFgSAP-2, rFgLAP+rFgSAP-2−vaccinated (combined group), alum adjuvant−infected control and non−immunized−infected control. Each point represents the antibody titers and number of worm recovery; ●, the antibody titers and the number of worm recovery of rFgLAP−vaccinated group; ♦, rFgSAP-2−vaccinated group; ▼, rFgLAP + SAP-2−vaccinated group; ■, alum adjuvant-infected control group; ▲, non-immunized-infected control group.

**Table 1 tropicalmed-08-00334-t001:** Worm recoveries and percentages of worm reduction in ICR mice immunized with rFgSAP-2 and rFgLAP and challenged with 15 *F. gigantica* metacercariae per mouse.

Groups	Mice	Treatments	Worm Recovery	Mean ± SEM	Reduction (%)
1. uninfected control	10	non-immunized and uninfected			
2. infected control	10	non-immunized and infected	5,7,6,3,6,8,7,8,5,4	5.9 ± 0.53	-
3. alum control	10	immunized with alum adjuvant and infected	6,8,4,3,5,7,6,2,8,8	5.7 ± 0.68	-
4. rFgSAP-2	10	50 µg of rFgSAP-2 plus alum adjuvant and infected	2,4,0,1,2,0,1,2,4,0	1.6 ± 0.48	72.9 ^a^*, 71.9 ^b^*
5. rFgLAP	10	50 µg of rFgLAP plus alum adjuvant and infected	0,4,1,3,1,1,2,1,3,3	1.9 ± 0.41	67.8 ^a^*, 66.7 ^b^*
6. rFgSAP-2 + rFgLAP	10	50 µg of rFgSAP-2 + rFgLAP plus alum adjuvant and infected	2,1,0,0,1,1,3,2,0,1	1.1 ± 0.31	81.4 ^a^*, 80.7 ^b^*

* Significant in worm reduction compared with control groups at *p*-value less than 0.05. ^a^ Percent reduction, compared with group 2. ^b^ Percent reduction, compared with group 3.

**Table 2 tropicalmed-08-00334-t002:** The gross pathological lesions in vaccinated group of livers were analyzed by scoring the lesion damage.

Groups	Mean Liver Damage Score ± SEM	Percent Reduction
1. uninfected control	-	
2. infected control	3.7 ± 0.26	-
3. alum control	3.5 ± 0.40	-
4. rFgSAP-2—alum	1.5 ± 0.37	59.46 ^a^*, 57.14 ^b^*
5. rFgLAP—alum	1.5 ± 0.31	59.46 ^a^*, 57.14 ^b^*
6. rFgSAP-2 + rFgLAP—alum	1.0 ± 0.26	72.97 ^a^*, 71.42 ^b^*

* Significant in reduction of liver gross pathological lesion when compared with control groups at *p*-value < 0.05. ^a^ Liver damage score, compared with group 2. ^b^ Liver damage score, compared with group 3.

## Data Availability

Not applicable.

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
