# Peer review of "Combination Vaccines of Fasciola gigantica Saposin-like Protein-2 and Leucine Aminopeptidase"

_tropicalmed, 2023, doi:10.3390/tropicalmed8070334_

Round 1

Reviewer 1 Report

The manuscript contains very important experimental results that may lead to an immunotherapy procedure for the control of fasiolosis in livestock, the science is solid and the methods and interpretation is sound; however, the manuscript is in need of some english editing and clarification of some passages the are confusing or misspelled; further details are included. 

Abstract

Some english editing is required there are some misspellings but most importantly the redaction may be improved for a better understanding.

Line 13 …involved… this line contains a very succinct sentence describing the SAP-2 and LAP, some more elaborate description of the subject of study would be nice

Lines 14-21 FgSAP and FgLAP acronyms are not described previously, although Fg, SAP and LAP are correctly described, the combination of those acronyms are not.

Line 18 …”fine granules” is a vague term for cytology, should include a mesure reference

Lines 15-22 a very confusing sequence of experiments describing the vaccination, confocal imaging and SAP/LAP expression all at once, I suggest make a more orderly redaction of the summarized data

Introduction

Some english editing is required, the redaction may be improved for a better understanding.

Line 35 …”inadvertently humans”… does it mean that animals are advertently parasitized?; definitive hosts including humans, don’t need intention to acquire fasciolosis.

Lines 35-36 Fasciolosis is cause for economical loss to livestock products such as milk and meat industries; on the other hand significant monetary loses are attributed to medical treatment to animals and humans. A very clear distinction should be made between medical economical loses and those attributed to the livestock production of food. 

M&M

Lines 106-111

The supernatant of the pellets was collected and 0.45 µm filtrated, but never got inside the Ni-NTA columns; nevertheless somehow the purified recombinant protein came out of it, something is missing here.

Lines 137-155 an indirect immunofluorescence procedure is described, but not even mentioned along the text; I thought a direct immunofluorescent method was used until I read this section; this is a critical experimental approach, it is necessary to specify it every time it is mentioned in all sections.

Line 179 never heard the term “percent protection” maybe the author refer to the protection index against fasciolosis after immunization in percentage.

Results & discussion

I don’t recall the PAGE procedure described in M&M section.

Line 310 Very hard to see those granules

The manuscript is in need of some english editing and clarification of some passages the are confusing or misspelled, details in reviewer's comments

Author Response

Dear Professor

Thank you very much for your kind consideration, comments and suggestions. All the suggested corrections / additions in the revised manuscript have been made and colored in blue. 

Sincerely yours,

Asst.Prof.Dr. Pornanan Kueakhai

Reviewer 2 Report

The present research activity “The combination vaccines of Fasciola gigantica saposin-like 2 protein-2 and leucine aminopeptidase” by Narin Changklungmoa and authors shows an effective approach for the potential of single and combined rFgSAP-2 and rFgLAP vaccines with Alum adjuvant against F. gigantica.  Although this work is attractive for readers, however, I would like to raise some points before it goes for further process.

The work shows protective efficiency of single and combined rFgSAP-2 and rFgLAP, but there is a lack of immunoregulatory activities of single rFgSAP-2 and rFgLAP or combined in this study. In the case of immunoregulatory activities of target proteins have already been carried out previously, the authors should discuss in detail either in the introduction or in the discussion section.

In section 2.3. How the 5 µm thickness of the parasite section were produced for Co-localization of FgLAP and FgSAP-2?

In discussion section, The authors evaluated the synergistic relationship between Th1 and Th2 immune responses on the bases of IgG1 and IgG2a antibodies. Cytokines released from immune cells play key roles in the regulation of the immune response. However, I did not see any in vitro confirmation of cytokines release in response to rFgLAP and rFgSAP-2. Furthermore, the present work lacks some in vitro and in vivo studies before reaching a conclusion that these proteins could be vaccine-candidate antigens.

In lines 318-321, the author mentioned the combined percentage protection of rFgSAP-2 319 and rFgLAP, while in the last paragraph of the discussion, the percentage reduction of gross pathological lesions should discuss according to Table 2.

Figure 4. is difficult to read. Should improve. 

N/A

Author Response

(The authors gave the same response as above.)

Reviewer 3 Report

In this study, the authors tested the vaccine potentials of recombinant FgSAP-2 and rFgLAP in alum against F. gigantica infection in mice. They showed that 80.7 to 81.4% percent reduction of worm burden animals were achieved in immunized animals, and that the vaccinated mice showed of liver damage reduction as compared with control groups.

As the authors mention, it appears that the combined rFgSAP-2 and rFgLAP immunization in FDA-approved alum, is a promising vaccine candidate against F. gigantica infections, and possibly fasciolosis caused by other species as well, in economic animals. I believe this deserves publication in Tropical Medicine and Infectious Disease.

Author Response

(The authors gave the same response as above.)

Round 2

Reviewer 2 Report

The authors of this Manuscript (2303542) have justified all the concerns and improved the MS. I recommend this MS for publication in TropicalMed MDPI after proofreading.

Author Response

Dear Professor

We would like to submit our rivised manuscript (tropicalmed-2303542) entitled “Combination vaccines of Fasciola gigantica saposin-like protein-2 and leucine aminopeptidase” to be considered for publication in Tropical Medicine and Infectious Disease. We thank you very much for your kind attention and consideration.

Sincerely yours,

Asst.Prof.Dr.Pornanan Kueakhai
